

# Palustrine forested wetland vegetation communities change across an elevation gradient, Washington State, USA

Nate Hough-Snee

Four Peaks Environmental Science and Data Solutions, Wenatchee, WA, USA

## ABSTRACT

**Background:** Forested wetlands support distinct vegetation and hydrology relative to upland forests and shrub-dominated or open water wetlands. Although forested wetland plant communities comprise unique habitats, these ecosystems' community structure is not well documented in the U.S. Pacific Northwest. Here I surveyed forested wetland vegetation to identify changes in community composition and structure across an elevation gradient that corresponds to flooding stress, asking: (1) How do forested wetland plant communities change across an elevation gradient that corresponds to flood frequency and duration? (2) At what relative elevations do different plant species occur within a wetland?

**Methods:** I measured overstory tree basal area and structure and understory vascular plant composition in three zones: wetland buffers (WB) adjacent to the wetland, an upper wetland (UW) extent, and a lower wetland (LW) extent, surveying individual trees' root collar elevation relative to the wetland ordinary high-water mark (OHWM). I estimated understory plant species abundance in sub-plots and surveyed these plots' height above the OHWM. I used non-metric multidimensional scaling ordination to identify patterns in vegetation communities relative to wetland elevation, and tested for compositional differences between the WB, UW and LW zones using PERMANOVA. I calculated overstory and understory indicator species for each wetland zone using indicator species analysis.

**Results:** Forest overstory composition changed across the elevation gradient, with broad-leaved trees occupying a distinct hydrologic niche in low-lying areas close to the OHWM. Conifer species occurred higher above the OHWM on drier microsites. *Pseudotsuga menziesii* (mean elevation = 0.881 m) and *Tsuga heterophylla* (mean elevation = 1.737 m) were overstory indicator species of the WB, while *Fraxinus latifolia* (mean elevation = 0.005 m) was an overstory indicator for the upper and lower wetland. Understory vegetation differed between zones and lower zones' indicator species were generally hydrophytic species with adaptations that allow them to tolerate flooding stress at lower elevations. Average elevations above the OHWM are reported for 19 overstory trees and 61 understory plant species.

By quantifying forested wetland plant species' affinities for different habitats across an inundation gradient, this study illustrates how rarely flooded, forested WB vegetation differs from frequently flooded, LW vegetation. Because common management applications, like restoring forested wetlands and managing wetland responses to forest harvest, are both predicated upon understanding how vegetation

Corresponding author
Nate Hough-Snee,
nhoughsnee@fourpeaksenv.com

relates to hydrology, these data on where different species might establish and persist along an inundation gradient may be useful in planning for anticipated forested wetland responses to restoration and disturbance.

# INTRODUCTION

Forested wetlands, also known as "forested swamps" (*Franklin & Dyrness, 1988*) and palustrine forested wetlands (*Cowardin et al., 1979*), are biologically diverse ecosystems that support unique plant communities. Within the U.S. Pacific Northwest these communities include upland trees, shrubs and herbs on elevated hummocks, and hydrophytic species that occur in low-lying areas with high water tables and/or periodic to frequent inundation (*Keogh, Keddy & Fraser, 1999*). While non-forested wetlands may include coniferous overstory trees at low abundance, Pacific Northwest forested wetlands are unique in that mixed coniferous and deciduous tree canopies often persist to old age (*Painter, 2009*) based on diverse microtopography available for tree seedling establishment and hydrophytic tree species' relatively plastic adaptations to wetland hydroperiods, anoxic soils, and overstory light environments (*Harrington, 1987*; *Ewing, 1996*; *Stolnack & Naiman, 2010*). Despite their unique composition, Pacific Northwest forested wetlands' vegetation structure, including species size and location, are poorly understood relative to upland forest ecosystems and non-forested wetlands (*Painter, 2009*; *Adamus, 2014*). Few studies exist in the Pacific Northwest that quantify where forested wetland plant species, an important component of wetland habitats, occur relative to hydrology or wetland elevation, a proxy for wetland hydroperiod (*Ewing, 1996*; *Hough-Snee et al., 2015b*). Only *Painter (2009)* has described forest structure and old tree size distributions in Pacific Northwest forested wetlands.

Addressing this gap in understanding where wetland plants occur relative to elevation, and how elevation corresponds to structure, may improve regional forested wetland conservation and restoration actions as forested wetland vegetation is naturally distributed across hydrologic gradients (*Brinson, 1993*; *Keogh, Keddy & Fraser, 1999*) and can be significantly altered by hydrologic modification (*Middleton & Souter, 2016*). For example, wetland restoration efforts intended to mitigate forested wetland loss often plant tree species at appropriate elevations relative to flooding so that plants successfully survive and grow and that restored wetlands' vegetation composition eventually resembles the composition of functional forested wetlands (*Bledsoe & Shear, 2000*). Accordingly, studies of how forested wetland plants relate to even coarse hydrologic indicators can improve the understanding of common forested wetland plant species' hydrologic niches. This fundamental information can inform wetland restoration and management that is
predicated upon understanding what plant species can reasonably occur at different elevations relative to soil inundation and surface water flooding.

Similarly, studies of where different plant species occur within forested wetlands can provide hypotheses for how natural resource management activities, like timber harvest, that impact wetland hydrology may impact forested wetland vegetation. Within Washington State forest practices in and around forested wetlands[1], including state-level buffer and harvest guidelines, are based on the best available scientific literature, which is limited in the Pacific Northwest ("Chapter 76.09 RCW: Forest Practices"; *Beckett et al., 2016*). Forested wetlands are managed under Washington State Forest Practice Rules ("Chapter 76.09 RCW: Forest Practices"; *Washington State Forest Practices Board, 1975*; *Washington State Department of Natural Resources, 2005*) to effectively result in "no net loss" of ecosystem functions and services. This mandate means that wetlands, including forested wetlands, should be managed around active forestry to maintain the processes that create diverse vegetation structure and habitats, transport material and energy through watersheds, and that contribute to downstream water quality, flow regulation and flood attenuation. However, watershed-scale logging alters forested wetland hydrology, often causing a rise in water tables, and concurrent changes in vegetation composition (*Timoney, Peterson & Wein, 1997*; *Batzer, Jackson & Mosner, 2000*) and tree growth (*Ewing, 1996*). Understanding at what elevations different species occur across a flooding gradient may allow for the development of hypotheses as to what species might be excluded from a given wetland by increased water levels associated with forest harvest.

Here I investigated forested wetland vegetation composition and structure across a hydrologic gradient asking two primary questions:

1. How do overstory forest composition and structure and understory forest composition change across an elevation gradient from high to low above the ordinary high-water mark (OHWM) within a palustrine forested wetland?

2. At what elevations relative to the OHWM are different plant species found within forested wetlands?

### Study site

The study site was Ash Wetland, a 4.6-ha palustrine forested wetland (*Cowardin et al., 1979*) located within 1,740-ha Pack Experimental Forest, a managed research forest in the Western Cascades Lowlands and Valleys EcoRegion near Eatonville, Washington, USA (Fig. 1). Ash Wetland has an average elevation of 281-m and is geographically isolated from surface flow (*Tiner, 2003*). The water table rises with autumn and winter rain and falls throughout the growing season into late summer, with water levels generally peaking in late winter to early spring. Plant and soil evapotranspiration often dry most of the wetland soil surface by late summer in dry years. Mean daily temperature and total precipitation were 9.8 °C and 118.36 cm from 1980 to 2010; during the year of the study (2009), mean daily temperature was 9.9 °C and total precipitation was 116 cm (PRISM Climate Group, Oregon State University, http://prism.oregonstate.edu).

---

[1] Washington State Forest Practice Rules define forested wetlands as "any wetland or portion thereof that has—or if the trees present were mature, would have—at least 30% canopy closure (*Washington State Department of Natural Resources, 2005*)…" from overstory trees.
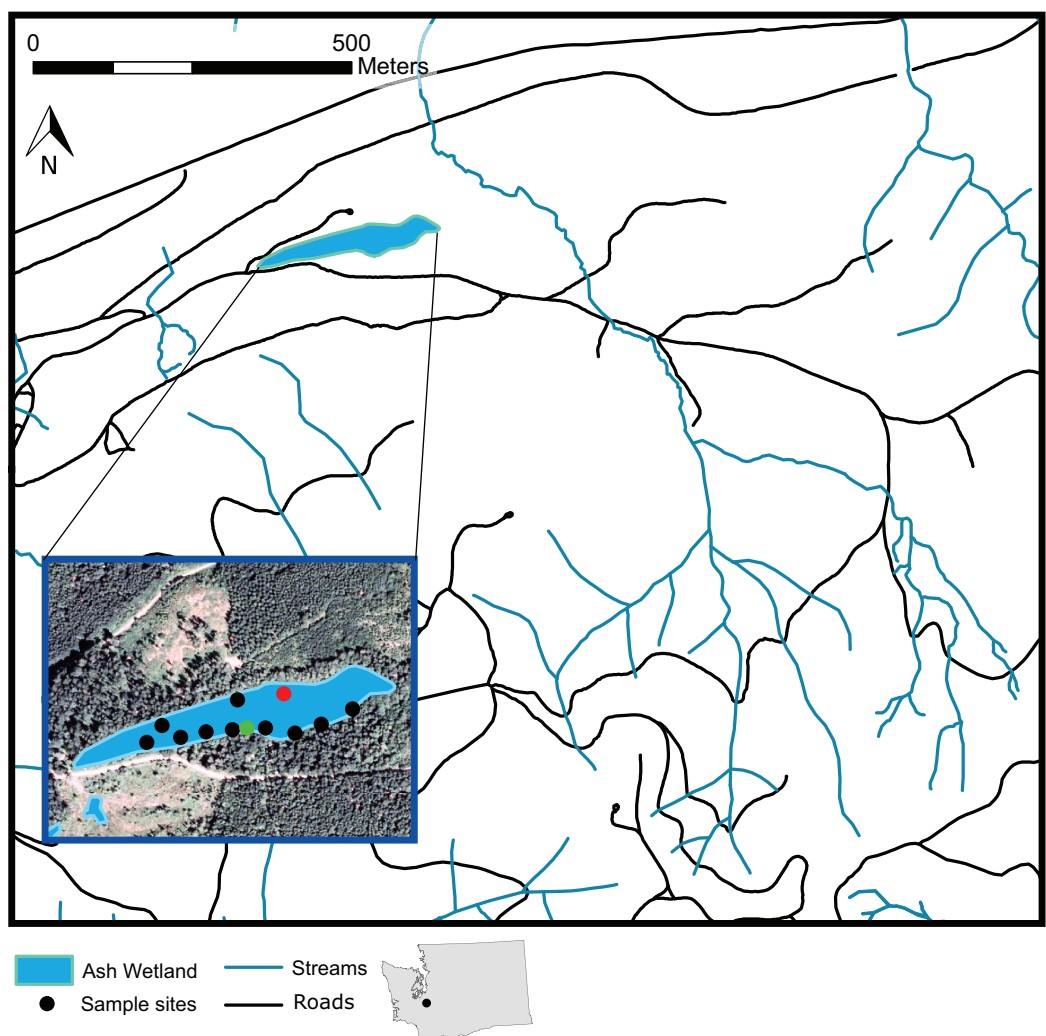

**Figure 1 A map of Ash Wetland and the sampled vegetation plots within Pack Experimental Forest, Eatonville, WA, USA.** Black dots indicate locations where full overstory and understory sampling occurred. Green dots indicate plots where only overstory vegetation was sampled. The red dot indicates a plot intended for sampling that could not be sampled.

# METHODS

I used the Army Corps of Engineers' 1987 Wetland Delineation Manual (*US Army Corps of Engineers Environmental Laboratory, 1987*) and Western Mountains, Valleys and Coasts regional supplement (*US Army Corps of Engineers, 2010*) to determine the wetland-upland boundary within Ash Wetland in September–November 2008, preliminarily surveying vegetation and assessing hydric soil and hydrology indicators along the wetland boundary. From this initial wetland delineation, I mapped the OHWM and identified three a priori zones across the wetland from which vegetation–elevation relationships were assessed: wetland buffer (WB), upper wetland (UW) and lower wetland (LW) (Table 1). These zones were based on within wetland elevation as it relates to the OHWM and used to stratify sampling as they mark breaks in inundation. The WB zone was the area immediately upslope from the OHWM and consisted almost entirely of
**Table 1 Wetland zones related to the ordinary high-water mark (OHWM) across which vegetation sampling was stratified.**

| Zone | Description |
|------|-------------|
| Wetland buffer (WB) | Upland area slightly above the OHWM. Soils were not hydric, and indicative of upland hydrology. Generally perched 0.2 m above the OHWM except where higher elevation microtopography existed |
| Upper wetland (UW) | The area that generally captured the OHWM and adjacent areas, including wetland soils, but generally inundated by <15 cm of water for a portion of the year |
| Lower wetland (LW) | Emergent, low wetland area below the OHWM, where standing water persists during a large portion of the growing season, often at depths >15 cm |

non-hydric, upland soils. The UW boundary began at the OHWM and ran across an elevation gradient into the wetland. UW soil pits were >70% hydric soil types, either histic epipedon or histosols and the LW was below the UW and characterized by hydric soils with aquic moisture regimes (Table 1). Sandy redoximorphic features, underlain by clay or rock restrictive layers, occurred in portions of the UW plots.

## Vegetation surveys

Within each wetland zone I sampled vegetation within overstory plots containing nested understory plots. Prior to sampling, I used GIS to overlay a 10-m grid to the wetland and randomly selected 12 plot locations adjacent to the wetland boundary at which sampling would occur in all three zones. These points were field verified as being on the wetland edge during the initial delineation, and 10 m × 10 m overstory forest plots within the WB, UW and LW, were oriented parallel to wetland slope. This sampling scheme effectively resulted in a stratified random sampling scheme with 12 sampling locations serving as blocks and three plots, one of each wetland zone (buffer, upper and lower) within each block (36 plots). One full set of overstory plots could not be sampled due to dangerous wildlife resulting in 33 total overstory plots (33 plots, 0.33 ha).

I identified all overstory trees >2.5 cm in diameter and >2 m in height within each overstory plot, measured tree diameter at breast height (DBH), and surveyed tree root collar elevation relative to the OHWM elevation. Understory species cover was estimated within ten 1 m × 1 m plots in each forest plot. All understory shrubs <2 m in height and <2.5 cm in DBH were included in understory vegetation cover and measured with herbs and forbs. Within the 33 plots sampled for overstory trees, one full set of understory plots and one LW plot had to be abandoned due to dangerous wildlife (ground nesting wasps) resulting in 290 understory plots. All plots were thought to be representative of Ash Wetland and so species rarefaction curves were not created to evaluate at what sampling intensity unique species numbers diminished.

## Elevation surveys

Base elevations of individual trees and understory plot centers were surveyed using a stadia rod and level and related to temporary benchmarks at the OHWM at each sampling location. I calculated height above the OHWM for each tree and plot by subtracting instrument height at each benchmark from rod height using standard land surveying
methods. Because survey measurements were not linked to benchmarks with known elevations, all vegetation elevations are relative to the OHWM elevation at that sample site.

## Statistical analyses

Individual tree species' mean elevation relative to the OHWM were compared using one-way ANOVA and generalized linear hypothesis testing by Tukey's pairwise multiple comparisons in the "mcp" function in the multcomp R package (*Hothorn, Bretz & Westfall, 2008*). This generalized linear hypothesis test approach was taken to test the hypothesis that elevation above the OHWM differed based on species while controlling for potential type one errors as described by *Hothorn, Bretz & Westfall (2008)* and *Bretz, Hothorn & Westfall (2011)*. Species DBH and elevation relative to the OHWM were plotted based on linear regression relationships to identify trends in tree DBH and wetland elevation. Because this was an ad hoc exploratory analysis, not a test of a mechanistic, causal relationship between tree size and elevation, formal statistical hypothesis testing was not used. Six species, *Spiraea douglasii*, *Taxus brevifolia*, *Abies grandis*, *Ilex aquifolium*, *Oemleria cerasiformis*, lacked sufficient replicates ($n > 2$) to assess their relationships between tree size and wetland elevation (Fig. 2).

Vegetation composition was compared across the elevation gradient using ordination methods, hypothesis testing and indicator species analysis (ISA). I converted DBH to basal area for each overstory tree and calculated each species' relative basal density and relative frequency, from which importance values (IV) were calculated for each overstory plot. Plot-level species IV were then used to calculate compositional dissimilarity between plots (Bray–Curtis distance) from which overstory forest composition was compared using non-metric multidimensional scaling (NMDS). NMDS ordination was also used to compare understory vegetation by zone based on Bray-Curtis distance. Overstory species IV and understory abundance values were regressed against each ordination solution to identify individual species relationships to community composition. Plot elevation was also regressed against the understory NMDS ordination.

I quantified differences in wetland zones' overstory and understory vegetation composition using PERMANOVA (*Anderson, 2001*; *Oksanen et al., 2019*; Table S1) and identified understory indicator species for each of the wetland zones using ISA, including multi-level pattern analysis for the understory and Dufrêne–Legendre ISA for the overstory (*Dufrêne & Legendre, 1997*; *De Caceres, Legendre & Moretti, 2010*). For all overstory community analyses the individual forest plots were the observational unit. For all understory community analyses, individual vegetation quadrats within each overstory plot were the observational unit and were stratified by wetland zone for both the PERMANOVA and ISA permutation tests. All analyses were performed using R statistical software (*R Core Team, 2018*). All statistical tests were performed with an alpha of $P < 0.05$.

## RESULTS

### Plant species elevations above the OHWM

I identified 19 overstory tree species and 61 understory plant species within the plots. Common conifer species within the plots, *Tsuga heterophylla*, *Pseudotsuga menziesii*

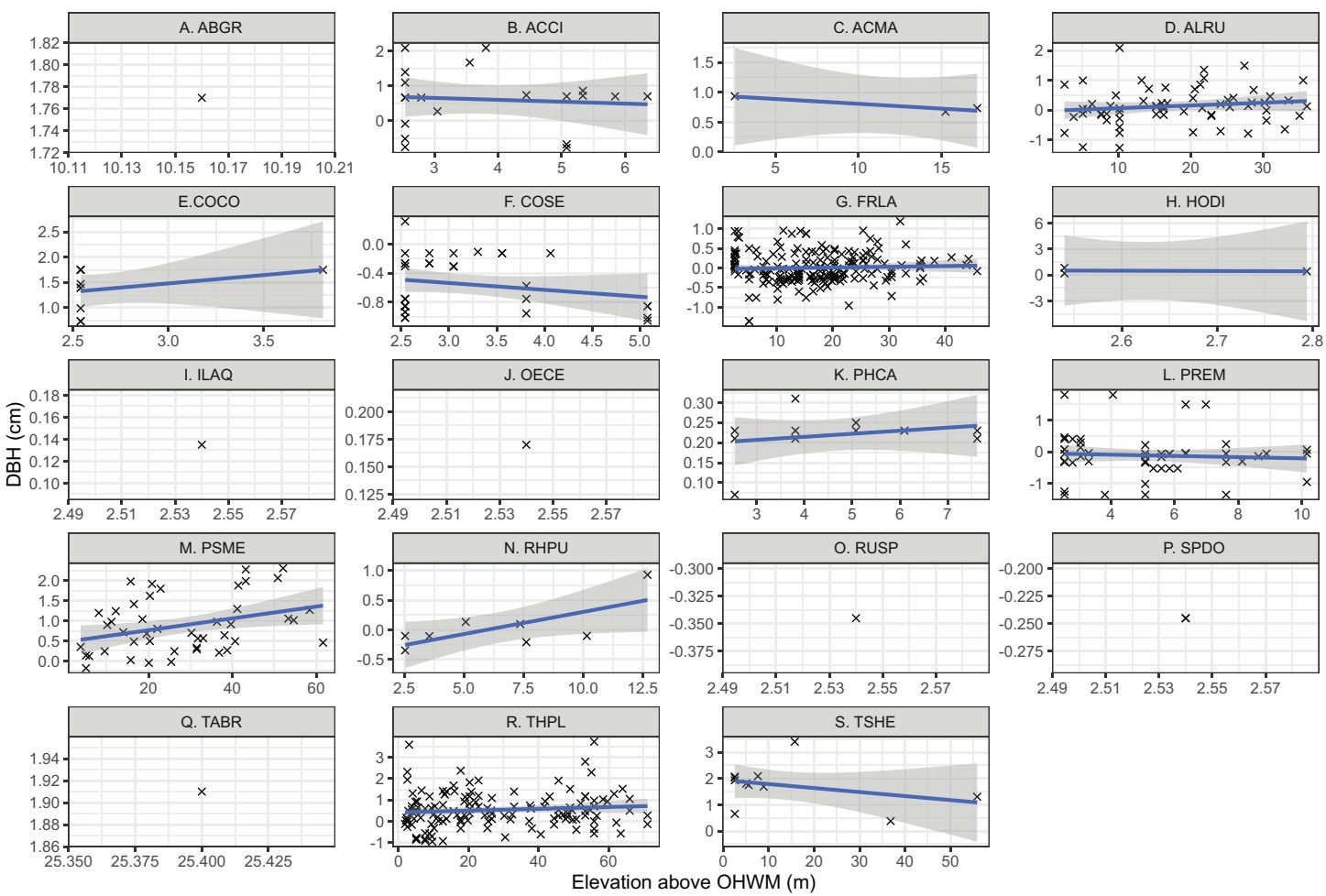

**Figure 2 Individual tree elevation above the ordinary high-water mark (OHWM) plotted against measured tree diameter at breast height (DBH) that was used to calculate estimated basal area.** Trees are plotted by species: (A) *Abies grandis*; (B) *Acer circinatum*; (C) *Acer macrophyllum*; (D) *Alnus rubra*; (E) *Corylus cornuta*; (F) *Cornus sericea*; (G) *Fraxinus latifolia*; (H) *Holodiscus discolor*; (I) *Ilex aquifolium*; (J) *Oemleria cerasiformis*; (K) *Physocarpus capitatus*; (L) *Prunus emarginata*; (M) *Pseudotsuga menziesii*; (N) *Rhamnus purshiana*; (O) *Rubus spectabilis*; (P) *Spirea douglasii*; (Q) *Taxus brevifolia*; (R) *Thuja plicata*; (S) *Tsuga heterophylla*. Trend lines are the linear regression relationship between elevation above OHWM and tree DBH, while point size reflects individual tree basal area ($m^2$). The shaded area is the 95% confidence interval for the regression relationship. Note that (C), (E) and (H) are low sample size observations and regression relationships have low confidence based on limited observations.

and the deciduous shrub, *Corylus cornuta* occurred at the highest surveyed elevations, roughly one meter or more above the OHWM (Figs. 2 and 3). *Thuja plicata*, the most common conifer species, occurred at a mean height of 0.53 m above the OHWM. Of the common deciduous, broad-leaved species, *Fraxinus latifolia*, *Rhamnus purshiana*, *Alnus rubra* and *Prunus emarginata* occurred near and slightly below the OHWM (Figs. 2 and 3). Most shrubs occurred within 25–50 cm of the OHWM, except for *Acer circinatum* which occurred 0.64 m above the OHWM. *Cornus sericea*, a hydrophytic shrub, occurred over 50 cm below the OHWM (Figs. 2 and 3).

For most tree species within the overstory, the relationship between elevation above the OHWM and DBH was positive (Fig. 2). That is, larger trees occurred higher above the most low-lying areas within the wetland. Both *T. heterophylla* and *C. sericea* size were
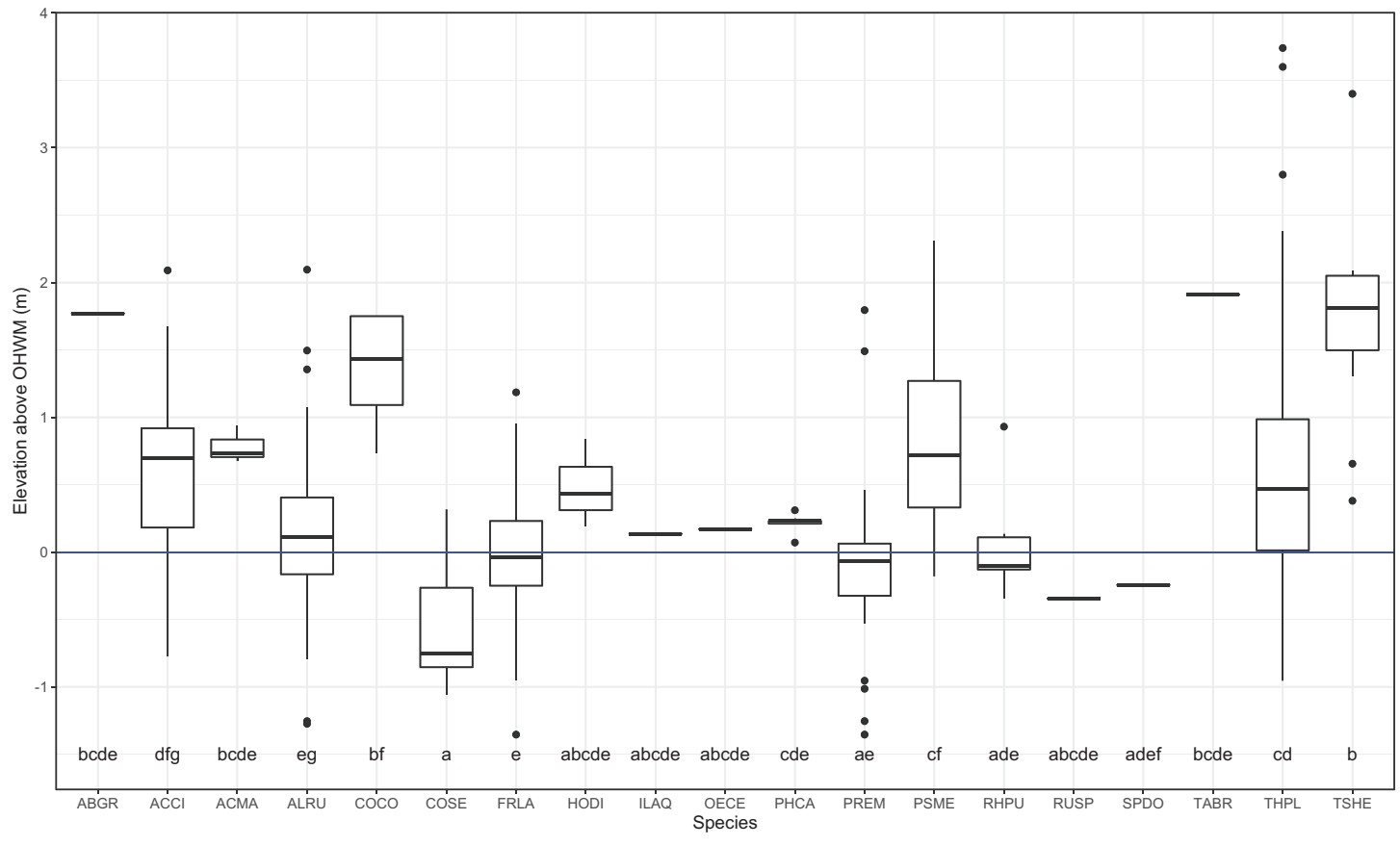

**Figure 3 Box and whisker plot of individual trees elevations above the ordinary high-water mark (OHWM) by species.** Different letter values indicate statistically significant differences in the mean elevation between species detected by Tukey pairwise comparisons (One-way ANOVA $F = 16.81$, $P = 2e^{-16}$). The bold line is the median and boxes are the 75th and 25th percentile of observations. Outlier values are any values over 1.5 times the interquartile range over the 75th percentile or any values under 1.5 times the interquartile range under the 25th percentile.

negatively correlated with wetland elevation (Fig. 2), meaning that larger individuals occurred in wetter, lower locations within the wetland.

## Overstory forest composition

I selected a three-dimensional NMDS ordination solution for overstory composition with an observed stress of 0.066 (non-metric fit $R^2 = 0.996$; linear fit $R^2 = 0.977$) and low probability of the final solution's stress being artificially low as an artifact of the data structure ($P = 0.040$; Monte Carlo randomization test). I also examined a scree plot of NMDS stress against NMDS axes and found that NMDS stress decreased from two to three axes, but only marginally decreased from three axes to four. This provided evidence for assessing community composition with the three-dimensional NMDS solution.

Of the 19 overstory species sampled, *F. latifolia* ($R^2 = 0.95$), *T. plicata* ($R^2 = 0.92$), *P. menziesii* ($R^2 = 0.88$), *A. rubra* ($R^2 = 0.77$), *A. circinatum* ($R^2 = 0.61$), *C. sericea* ($R^2 = 0.38$) and *P. emarginata* ($R^2 = 0.28$) were all significantly correlated to the final ordination solution at the $P = 0.05$ level (Fig. 4). *F. latifolia* was positively correlated to the

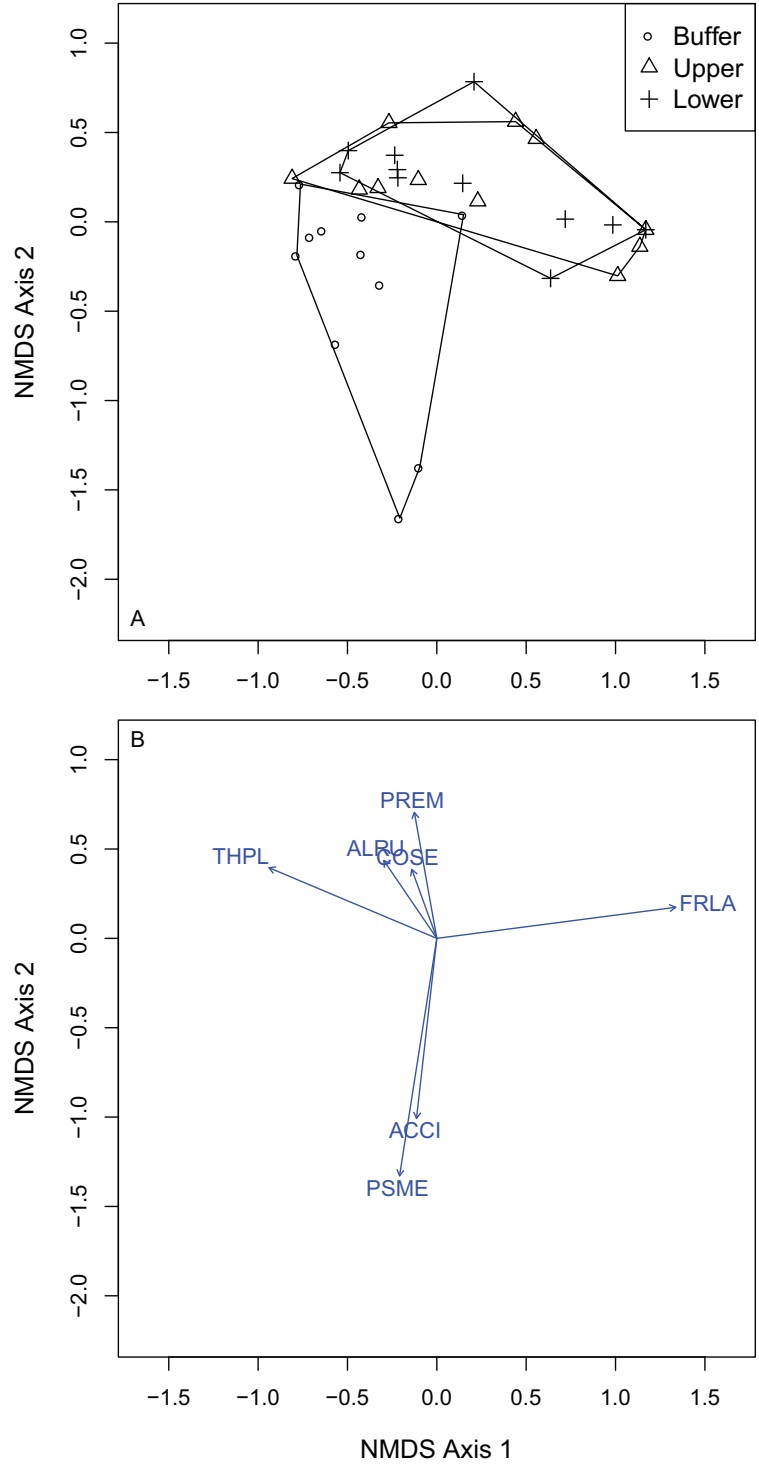

**Figure 4 NMDS ordination of the overstory vegetation showed that the upper and lower wetland plots differed in composition from the wetland buffer, but not from one another and that vegetation differed across both axes reflective of hydrologic gradients.** (A) Overstory plots by treatment: buffer, upper and lower wetland. (B) Vectors indicate species that were significantly correlated to the final NMDS solution at the alpha of $P = 0.05$. Species codes correspond to those in Table 3.

**Table 2 Overstory tree indicator values derived from indicator species analysis.**

| Species | Code | Zone | Indicator value | P | Mean height above OHWM (meters) | Wetland indicator status |
|---|---|---|---|---|---|---|
| *Pseudotsuga menziesii* | PSME | Wetland buffer | 93.9 | 0.005 | 0.881 | FACU |
| *Tsuga heterophylla* | TSHE | Wetland buffer | 55.3 | 0.025 | 1.737 | FACU |
| *Fraxinus latifolia* | FRLA | Upper–Lower wetland | 81.6 | 0.035 | 0.005 | FACW |

**Note:**
Wetland indicator status is from the 2016 U.S. National Wetland Plant List (*Lichvar et al., 2016*): OBL, obligate wetland; FACW, facultative wetland; FAC, facultative; UPL, upland. For the full overstory indicator species list, see Table S3.

first and second NMDS axes (Fig. 4; Table S2), across both of which the gradient from UW to LW showed compositional differences. *P. menziesii* and *A. circinatum* were strongly correlated to negative scores across the second NMDS axis, where WB plots were most common (Fig. 4). Based on the elevations of individual species, positive to negative values across the first and second NMDS axes could be interpreted as low and wet plots to high and dry plots.

Overstory forest composition differed between the WB, UW and LW plots (PERMANOVA $R^2 = 0.25$; $P = 0.0001$; Table S1). Pairwise comparisons indicated that WB overstory composition significantly differed from that of the UW (PERMANOVA $R^2 = 0.22$; $P = 0.0001$) and LW plot composition (PERMANOVA $R^2 = 0.23$; $P = 0.0008$). The upper and LW plots did not significantly differ in their overstory composition (PERMANOVA $R^2 = 0.02$; $P = 0.82$). Because there was no difference between the upper and LW plots, I performed Dufrene–Legendre ISA between the combined upper and lower wetland plots and the buffer plots. ISA found that *P. menziesii* (indicator value = 93.9; $P = 0.005$) and *T. heterophylla* (indicator value = 55.3; $P = 0.045$) were indicator species for the WB and *F. latifolia* (indicator value = 81.6; $P = 0.03$) was the only significant indicator for the combined lower and UW zones (Table 2; Table S3).

## Understory composition

I selected a three-dimensional NMDS ordination solution for understory composition with an observed stress of 0.146 (non-metric fit $R^2 = 0.979$; linear fit $R^2 = 0.868$). Understory plot distance above the OHWM was significantly positively associated with the first NMDS axis ($R^2 = 0.161$; $P = 0.001$). Both plot elevation above the OHWM and vegetation composition changed across the first and second axes within the ordination. The second NMDS axis ran from high (dry) to low (wet) from positive to negative values. The first NMDS axis ran from high (dry) to low (wet) from negative to positive values. There were 24 plant species that were significantly correlated with the final NMDS solution at the $P = 0.05$ level (Fig. 5; Table S2). *Carex obnupta*, an obligate wetland species, was strongly associated with deeper, wetter habitats ($R^2 = 0.481$) while in contrast, *Gaultheria shallon* ($R^2 = 0.555$) and *Polystichum munitum* ($R^2 = 0.488$) were more strongly associated with drier, higher habitats. Generally, plants with affinities or tolerances for flooding

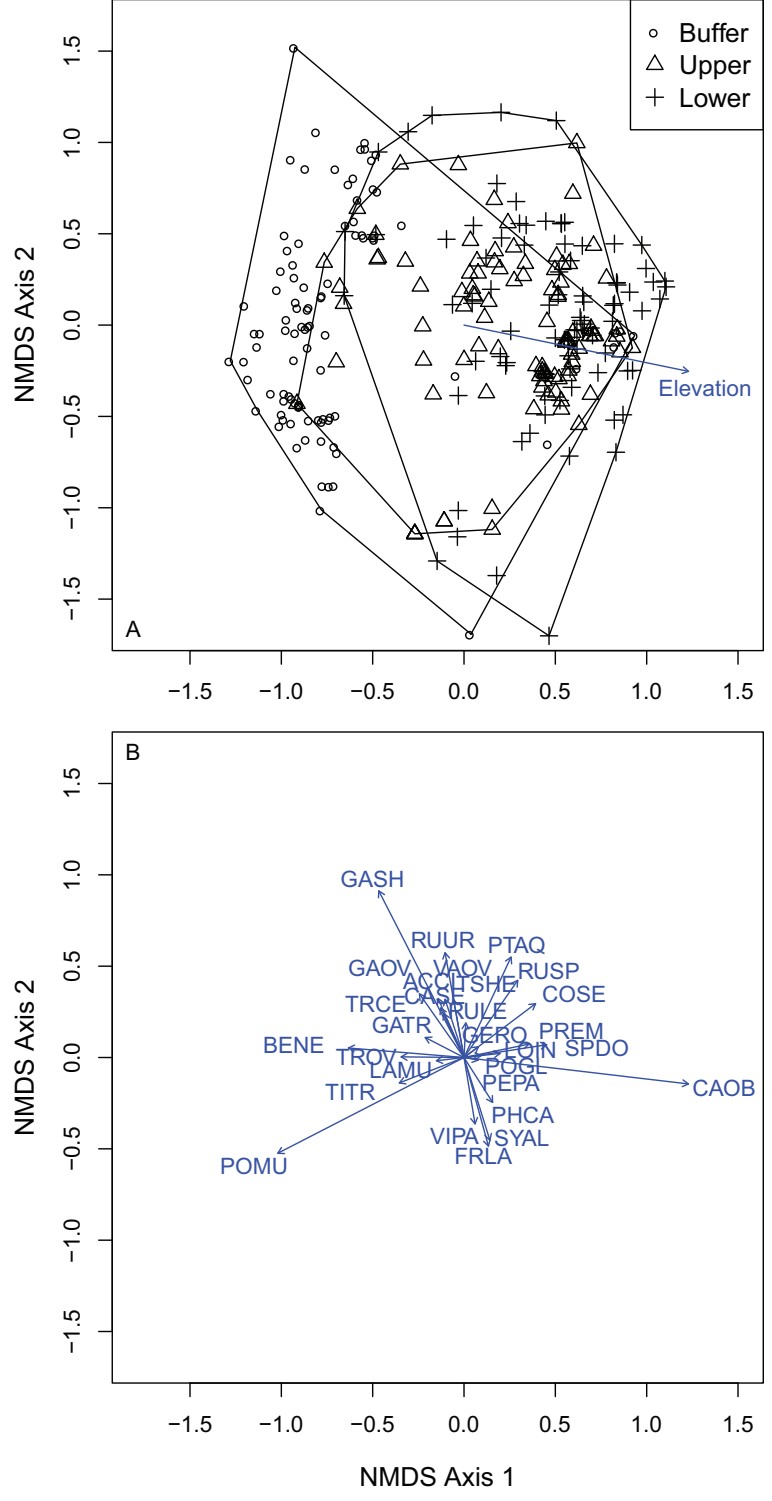

**Figure 5 NMDS ordination of the understory plots showed that vegetation composition was distributed across a gradient from high elevations to low elevations.** (A) Understory plots by treatment: buffer, upper and lower wetland. (B) Vectors indicate species that were significantly correlated to the final NMDS solution at the alpha of *P* = 0.05. Species codes correspond to those in Table 3.

**Table 3 Understory community indicator values derived from indicator species analysis (multi-level pattern analysis).**

| Species | Four-letter code | Zone | Indicator value | Probability | Mean height above OHWM (Meters) | Wetland indicator status |
|---|---|---|---|---|---|---|
| *Polystichum munitum* | POMU | WB | 74.7 | 0.005 | 0.2978 | FACU |
| *Mahonia nervosa* | BENE | WB | 52.9 | 0.005 | 0.6095 | FACU |
| *Acer circinatum* | ACCI | WB | 25.6 | 0.015 | 0.6422 | FAC |
| *Gaultheria ovatifolia* | GAOV | WB | 24.5 | 0.020 | 1.2367 | FAC |
| *Petasites frigidus* ssp. *palmatus* | PEPA | UW | 20.0 | 0.04 | −0.6970 | FACW |
| *Carex obnupta* | CAOB | LW–UW | 86.9 | 0.005 | −0.4707 | OBL |
| *Pteridium aquilinum* | PTAQ | LW–UW | 45.9 | 0.005 | −0.1343 | FACU |
| *Spiraea douglasii* | SPDO | LW–UW | 43.0 | 0.005 | −0.4968 | FACW |
| *Matricaria discoidea* | MADI | LW–UW | 38.1 | 0.045 | −0.2779 | FACU |
| *Rubus spectabilis* | RUSP | LW–UW | 36.3 | 0.005 | −0.3948 | FAC |
| *Symphoricarpos albus* | SYAL | LW | 38.2 | 0.005 | −0.9969 | FACU |
| *Cornus sericea* | COSE | LW | 37.6 | 0.005 | −0.2478 | FACW |
| *Prunus emarginata* | PREM | LW | 33.5 | 0.020 | −0.1860 | FACU |
| *Physocarpus capitatus* | PHCA | LW | 24.4 | 0.005 | −0.6970 | FACW |
| *Rosa nutkana* | RONU | LW | 23.6 | 0.005 | −0.8680 | FAC |
| *Amelanchier alnifolia* | AMAL | LW | 18.3 | 0.045 | −0.1733 | FACU |
| *Rubus ursinus* | RUUR | WB-LW | 35.6 | 0.02 | −0.1626 | FACU |

**Note:**
WB = Wetland buffer, UW = Upper wetland, LW = Lower wetland. Wetland indicator status is from the 2016 U.S. National Wetland Plant List (*Lichvar et al., 2016*). For a full understory indicator species list, see Table S3.

occurred along the wet side of the ordination axes. Many plant species were weakly but significantly associated with the final ordination solution (Table S2).

Understory composition differed among all of the wetland elevation zones (Table S1). Because composition differed between all treatment zones, I used multi-level pattern ISA (*De Caceres, Legendre & Moretti, 2010*) to identify where species were indicators of multiple zones. The WB zone had four significant indicator species: *P. munitum*, *Mahonia nervosa*, *A. circinatum* and *Gautheria ovatifolia* (Table 3). *Petasites frigidus* was the only significant understory UW indicator species. There were six significant LW indicator species: *Symphoricarpos albus*, *C. sericea*, *P. emarginata*, *Physocarpus capitatus*, *Rosa nutkana*, *Amelanchier alnifolia*. There were five significant indicator species of both the upper and lower wetland: *C. obnupta*, *Pteridium aquilinum*, *Spiraea douglasii*, *Matricaria discoidea*, *Rubus spectabilis*. *Rubus ursinus* was the only significant indicator species for both the WB and LW.

## DISCUSSION

Here I quantified how vegetation changes across an elevation gradient (question one) and the elevations at which overstory and understory vascular plant species occurred within a palustrine forested wetland (question two). Because wetland elevation corresponds to the frequency, duration and depth of flooding and soil saturation at a given location, pairing species and elevation has numerous applications. Within a wetland, elevation

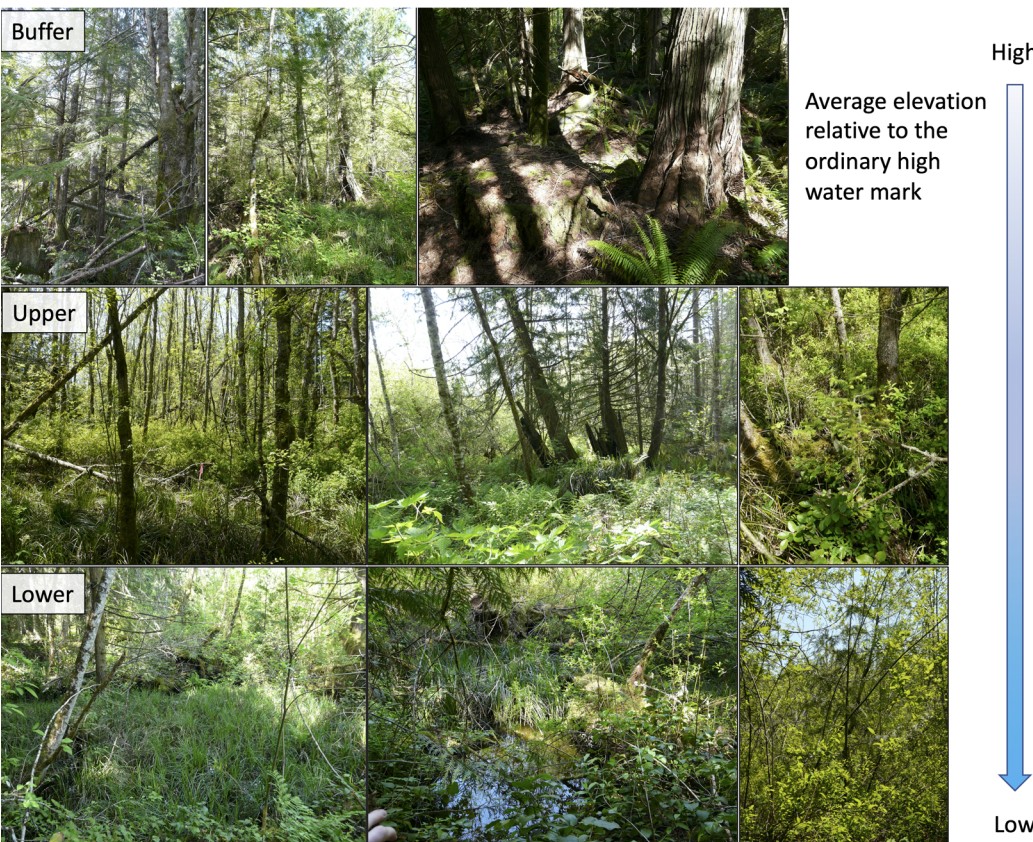

**Figure 6 Examples of forest plots from high to low across the wetland buffer, upper wetland and lower wetland groups.** Note overstory indicator species *Pseudotsuga menziesii* and *Tsuga heterophylla* alongside understory indicator *Polystichum munitum* in the wetland buffer row and upper and lower wetland overstory indicator species *Fraxinus latifolia* alongside understory upper and lower wetland indicator species *Carex obnupta* in the other rows.

from high to low dictates where plant species can establish and survive across a flooding stress gradient that excludes species without sufficient adaptations to flooding (e.g., aerenchyma, adventitious roots, etc.; *Keddy & Ellis, 1985*; *Battaglia, Collins & Sharitz, 2004*). Accordingly, the elevation gradient from high to low across which flooding stress increases is a measurable predictor of wetland ecosystem composition, including soil chemistry (*Yu & Ehrenfeld, 2010*), plant species (*Seabloom & Van Der Valk, 2003*), invertebrates (*Gathman & Burton, 2011*) and microbes (*Ahn et al., 2009*). The primary finding presented here, that forested wetland vegetation composition shifts from generalist, upland species at high elevations to more specialist wetland species at low elevations, aligns with these well-documented studies of how wetland elevation controls ecosystem processes.

Upland conifer species *P. menziesii* and *T. heterophylla* were the primary indicator species of the WB and occurred one meter or more above the OHWM relative to deciduous species like the upper and lower wetland indicator *F. latifolia* (Fig. 6), which occurred roughly at the OHWM. For the most abundant tree species, *A. rubra*, *F. latifolia*, *T. plicata* and *P. menziesii*, DBH was inversely correlated with depth above OHWM.

This finding is consistent with studies elsewhere that show more deeply and frequently flooded trees incur flood-induced physiological stress that may impede growth or survival relative to trees at higher elevations or lower flooding levels (*Ewing, 1996*; *Walls, Wardrop & Brooks, 2005*). Other studies have shown that wetland vegetation composition and structure changes across flood-stress gradients from low to high or more frequently to less frequently inundated (*Battaglia & Sharitz, 2006*; *Gathman & Burton, 2011*; *Berthelot et al., 2015*).

The negative relationship between tree elevation and tree size for most species can be interpreted one of two ways: flooded, (1) non-wetland trees are physiologically stressed and grow more slowly in areas of frequent inundation and high flooding stress, or (2) environmental conditions have changed as vegetation succession occurred and/or natural interannual hydrology varied, allowing for the recent establishment of younger trees in certain microsites–hydrophytic species in low, wet areas and upland species on fallen wood or stumps. While either or both of these patterns are plausible, tree age was not measured alongside tree size, making it difficult to decouple the causal mechanisms behind these observations.

Several of the observed elevation differences between overstory species may be explained by individual plant species' traits that allow them to persist in flooded conditions. For example, *F. latifolia*, which occurred close to the OHWM and is also a facultative wetland species (FACW; *Lichvar et al. 2016*), blooms late and drops seeds after peak floods have receded, a strategy that avoids flooding (*Lenssen, Van de Steeg & De Kroon, 2004*). Previous research suggests that *A. rubra*, a facultative wetland plant, is more sensitive to flooding than *F. latifolia* (*Ewing, 1996*), but here I found no significant difference in the elevations at which overstory trees of both species occurred.

Within the understory, *A. rubra* occurred at lower elevations than *F. latifolia*. This may be attributable to *A. rubra*'s dense seed rain and relatively fast growth rate, which allows seedlings that establish to grow quickly enough to spread their roots to higher adjacent hummocks and other landforms. *C. obnupta*, a rhizomatous and aerenchyma-dense obligate wetland plant, was an indicator of both the upper and lower wetland, which is consistent with a greenhouse study that showed the species to be physiologically resilient to different flooding regimes (*Hough-Snee, 2010*; *Hough-Snee et al., 2015b*). The long-term survival and growth mechanisms for many of the overstory and rhizomatous understory species observed within Ash Wetland may be different than the short-term establishment and survival mechanisms examined in studies of smaller seedlings and saplings (*Ewing, 1996*).

## Applications to wetland management and future directions

The data presented here illustrates where forested wetland plant species occur relative to flooding (OHWM), and this information can be used to place species into appropriate hydrologic context when anticipating wetland change from hydrology altering management activities, like forested wetland and/or watershed timber harvest. While the observed relationships provide insight into the natural history of Ash Wetland and similar palustrine forested wetlands, these relationships also have implications for the region-wide

management of palustrine forested wetlands. Watershed- and harvest unit-scale timber harvest, roads, and other land management that raises water tables or increases the duration and magnitude of flooding will likely shift forest composition toward hydrophytic stress tolerant species (*Devito, Creed & Fraser, 2005*; *Houlahan et al., 2006*).

In contrast, if forested wetlands are ditched or drained to facilitate forest harvest, then flood-tolerant, hydrophytic species may be encroached upon by shade-tolerant upland species. These hypotheses have not been tested within forested wetlands in Washington State, and any such characterization of forested wetland dynamics over time in response to hydrologic modification would immediately inform forested wetland management around industrial forests (*Adamus, 2014*; *Beckett et al., 2016*). Since forested wetland vegetation provides foundational habitat used by birds (*Cooke & Zack, 2008*) and mediates hydrological processes that contribute to downstream aquatic habitats (*Richardson, 2012*), quantifying how forested wetland vegetation may change in response to altered disturbance and hydrologic regimes is a research priority that will directly inform biodiversity conservation in the Pacific Northwest and beyond.

This study provides context into where plant species occur along an elevation gradient that reflects wetland hydrology within an isolated forested wetland. While the relationships between species elevations and the OHWM are informative, the data presented here do not identify the specific mechanisms that allow some species to occur at a given location within the wetland and while other species are precluded from occurring. For example, I used a coarse hydrologic indicator (OHWM) to map the lateral hydrologic extent of a wetland, rather than measuring hydrologic regimes over time. Forest managers often wish to know how harvest will change hydrology at the stand to sub-basin scales and then how this change in hydrology will alter forest composition over time. This study does not identify whether a given tree or species established or matured during a wet or dry period or where the tree established relative to peak hydrology in the year of establishment, but instead provides evidence of where species occur relative to flooding stress.

Future regional investigations in forested wetland ecology should focus on how biological, physiological, and hydrological attributes of these unique ecosystems intersect to shape forest composition over multiple timeframes, including when and how forest species establish and grow relative to frequent, low-magnitude flooding and infrequent, high-magnitude flooding and/or drought. While interspecific patterns between species are explained here, intraspecific trait diversity shapes species' capacity to tolerate stress, compete, and reproduce in flooded environments (*Hough-Snee et al., 2015b*), with implications for how wetland plant communities assemble (*Hough-Snee et al., 2015a*). Because species' genetics limit the range of traits that allow species to establish and persist amid wetland hydrologic and biophysical stressors (*Lenssen et al., 2004*), intraspecific variability in species' adaptations to flooding should also be considered when comparing spatially disparate wetlands that hold the same species.

Improving the body of knowledge around where different wetland species occur within wetlands also has applications to restoration planning. Restoration practitioners can assimilate species–elevation relationships into wetland restoration plans by designing wetland planting gradients to ensure that the most appropriate species are planted at a

given location (e.g. hydrologic niche or elevation) within a wetland. Additionally, potential state and transition models can be created for different forested wetland communities where vegetation may change over time as wetland hydrology becomes wetter or drier from disturbance, restoration, or vegetation succession.

## CONCLUSION

This study characterized the relationship between forested wetland plant species and relative elevation above the OHWM, a proxy for the hydrologic extent of a wetland. I quantified the ranges of elevations across which species with adaptations to wetland conditions were more likely to occur. Deciduous shrubs and trees occurred at lower elevations within the wetland and had higher measured DBHs within flooded environments than upland species that lacked adaptations to flooding. These results enumerate ecohydrological species–elevation relationships within a Pacific Northwest palustrine forested wetland, relationships that illustrate patterns of how different plant species are distributed relative to flooding stress. Additionally, this study provides rare, regionally relevant observational data, a starting point from which future hypotheses can be mechanistically tested to understand how different plant species establish, grow and persist within forested wetlands under different hydrologic regimes.

## ACKNOWLEDGEMENTS

Derrick Cooper, Lexine Long, and Marco Negovschi were invaluable field helpers during this project. Rodney Pond and Drs. Greg Ettl, Soo-Hyung Kim, and Lloyd Nackley were helpful during experimental design. I am especially grateful to Dr. Kern Ewing (retired) for his mentoring and advice during the M.S. degree from which this project originated. I greatly appreciate his friendship and advice over the last decade.

### Funding
Nate Hough-Snee received funding from the Society of Wetland Scientists' Pacific Northwest Chapter to present this work at Society of Wetland Scientists meetings. The Center for Sustainable Forestry at Pack Forest provided housing and site access during fieldwork. The funders had no role in study design, data collection and analysis, decision to publish, or preparation of the manuscript.

### Grant Disclosures
The following grant information was disclosed by the authors:
Society of Wetland Scientists' Pacific Northwest Chapter.

### Competing Interests
Nate Hough-Snee is an employee of Four Peaks Environmental Science and Data Solutions.

## Author Contributions

- Nate Hough-Snee conceived and designed the experiments, performed the experiments, analyzed the data, prepared figures and/or tables, authored or reviewed drafts of the paper, and approved the final draft.

## Data Availability

Data is available at Figshare: Hough-Snee, Nate (2019): Data from PeerJ Submission: How do forested wetland plant community and species distributions relate to flooding stress? A case from a palustrine forested wetland, Washington State, USA. figshare. Dataset. DOI 10.6084/m9.figshare.10048349.v2.

## Supplemental Information

Supplemental information for this article can be found online at http://dx.doi.org/10.7717/peerj.8903#supplemental-information.

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
