# Peer review of "Palustrine forested wetland vegetation communities change across an elevation gradient, Washington State, USA"

_PeerJ, doi:10.7717/peerj.8903_

## Round 0.1 · original submission · Major Revisions

Dear Nate Hough-Snee,

I have now received two reviews for your manuscript: “How do forested wetland plant community and species distributions relate to flooding stress? A case from a palustrine forested wetland, Washington State, USA. ”. Both reviewers are positive about your manuscript but suggest some changes before acceptance. Most of these changes are relatively minor (see also my comments below), but please pay particular attention in answering the points raised by reviewer 2 concerning the experimental design. Because, these points might have an important impact on the manuscript, it need major revision before possible acceptance.
I look forward receiving a revised version of your manuscript.

Sincerely

Yann Salmon


Additional comments:

- question 2 should be better introduced

L183 I think the Figure 3 is the relevant one here (not 2). I suggest to present the results as boxplot rather than barplot of the mean to give an idea of the distribution

L192 to illustrate this point it would make more sense to switch the axes of fig. 2

L327-335 being honest about the study limitation is good. However, it would help (especially if it is to inform forest managers) to explain how much these limitation impact the findings and their interpretation.

In relation to reviewer 1 comment, the beginning of the conclusion is more a summary than a conclusion

Fig. 2 a lot of species have very few data point and the regression seem very sensitive to some extreme values (e.g. RHPU).

Fig. 3 see comments above

Fig. 4 and 5, using color would make the figure easier to read

·

Basic reporting

The author has used professional English throughout the paper. It is well written and easy to read. The author is well versed with the topic.

In the Results section of the Abstract (line 41 – 47) “This study quantifies…” is more like the aims and objectives of the study and should be stated there. It could be replaced by the research questions in the Background section of the Abstract.

Generally the introduction and background lacked structure in the sense it does not give a broad scope of the study and its relevance. For example, what are some of the “ecosystem functions and services” provided by these wetland vegetation? The last paragraph of the introduction and background (lines 84 – 90) could be expanded a bit more where the points 1 and 2 currently stated as a question are structured as the objectives of the study clearly stating the purpose and techniques used to address the aim of the study in lines 84 – 85.

Figures are relevant but bot great in quality. I suggest the following changes;
- Figure 1 – useful to include the names of some of the major streams within the catchment. Contour lines may also be useful to show elevation.
- Figure 2 – graphs need to be made clearer. Could DistWater be changed to different symbols rather than a colour gradient? Increase font size of the axis labels
- Figure 3 – the bar graph does not need a colour gradient, the bars itself indicate the values. What are the codes on the bars and on the x-axis? It may be helpful to include the scientific names of the trees on the bars rather than the codes which have not been defined anywhere in the paper.
- Graphs in Figures 4 should have individual labels, for example could be labelled as Figure 4a and Figure 4b and referred to accordingly in the paper. Same goes for the graphs in Figure 5.
- Figure 5 graphs – ensure that the labels are not overlapping so that they can be read clearly

Appendix 1 and 2 referred to in the context of the paper were not included in the paper though mentioned several times throughout the document

Experimental design

The title indicates that the paper is about flooding stress on freshwater wetland vegetation however this message did not come out prominently in the findings of this study. While it is addressed throughout the paper, the findings should be emphasised in greater detail. The author has stated that this study “will improve regional understanding of forested wetland vegetation and inform restoration and forest practice policies” but does not provide clear signposts of the in the article. Also, this is not later addressed in the conclusions. I therefore recommend that before the paper is published, the overall significance of the findings needs to be elaborated and emphasised more clearly in the discussion and conclusions of this article.

I suggestion the following changes to the discussion section to focus on the findings;
- First paragraph of discussion is background information. The author should start their discussion with a sentence like that in line 270. Lines 313 – 325 is also background information and should not be in the discussion. These could be moved to the introduction section
- Line 271 and 272 is missing how many “meters above the OHWM relative to deciduous species… “
- Line 282 “Because I did not measure tree age…” is not a good way to start a paragraph. Also check this paragraph – the two points made, are they related to flooding?
- Discussion could be divided into sub-headings addressing the different objectives of this study. Section 1 – “overstory and understory forest composition change across an elevation gradient”, discuss how similar or different it is from other sites. Section 2 – “what elevations relative to the ordinary high-water mark are different plant species found within forested wetlands” and discuss which vegetation has a higher capacity to tolerate flooding. The final 2 paragraphs of the discussion should be a separate section called “limitations and recommendations”

In the Methods section, the author could expand on the threshold of the statistical analysis performed; “All statistical tests were performed with an alpha of P < 0.05” which means that the relationship was deemed statistically significant between the measured variables.

Validity of the findings

In my opinion all underlying data have been provided; they are robust, statistically sound, & controlled but a few points could be made clearer. For example;

- What percentage of “species lacked sufficient replicates”
- Sentence in lines 206 – 207 and 224 – 226 should be in your methods as it was the techniques used for the different scenarios.

The conclusion of this article is a bit weak and could be improved by incorporating the suggestions I made earlier. The author could include names of plants that most prominently showed resilience to flooding to those that did not. Sentences, such as that in lines 354 – 356 should not be in your conclusion.

Additional comments

I recommend stylistic changes to the paper, for example;
- The overuse of personal pronouns gives the article an informal tone and makes it too subjective. I would suggest making changes where “I” could be replaced by statements like - “This study aims to investigate…” and “The overstory tree and understory vascular plant composition and structure of the three zones was measured using…” and “This study was done by collecting…”
- Paragraphs or sentences should not begin with the conjunction “because”
- Very long sentence for example, lines 54 – 59 that should be separated.
- Check sentences in lines 72, 97, 100 – 102, 113, 149, 211, 307 – 310.
- Explain/define “exploratory analysis”
- Subheadings in line 123, 137, 145 should be secondary to Methods sub-heading
- FACU not defined in the article
- Species names of the different vegetation tend to be long and sometimes difficult to include in graphs, they therefore should be properly abbreviated and used consistently throughout the paper.

·

Basic reporting

The present manuscript investigates species composition of overstory and understory plants along an elevational gradient in a forested wetland in the U.S. NW Pacific. Overall, I judge the manuscript design, style, and presentation as suitable, however, I think that the methodology presents some aspects/uncertainities which should be improved before acceptance (details below). In addition, I have suggestions for some minor issues which also could be improved (details below).

Experimental design

The entire study design, main results, and discussion is dependent on the exact location of plant species relative to the ordinary high water mark (OHWM) and the delineation of the three elevational units (lower and upper wetland, wetland buffer). Therefore, I think that the definition of the OHWM and the three zones should receive more attention, also concerning potential bias. The OHWM is variable in time and scale, even in such a small forest swamp. Therefore, mean days of inundation per time unit would provide the best scale for vegetation inventories along the elevational gradient. In my point of view, it is not sufficient to describe the different soil types for each zone for exact delineation, rather, exact measurements (i.e., by using a theodolite or any other kind of measuring device) relative to the water level should be provided. If soil types were the unique fator evaluated leading to the definition of the three elevational units, please indicate how many soil samples, profile descriptions etc. were performed.

In L125ff you state a 10m grid of your wetland for the selection of vegetation plots - but this is not shown in Fig.1. I suggest to present the GIS-based map in this figure, which also could show the three elevational units that were mapped out.

Starting in L128 you mention the 10x10m plots for vegetation sampling, but it is unclear how many plots were established in total, and in each unit WB, LW, and LW. As I understood, you did 12 x 10x10 m plots = 1200 m2 in each unit (0,36 ha in total). Correct? Please clarify and state like this.

I L133, you state that understory species cover was measured. How? Did you count the abundance and/or estimated the coverage? Especially for herbaceous species this would be an important information. How many individuals in both the understory and overstory were sampled in total?

In the statistical analyses, I missed a species rarefaction and/or species-area curve to test for representativity of your vegetation/species samples. Given the sampled area, is your inventory representative for the different vegetation types in both the understory and overstory?

Validity of the findings

L196: If species are not abundant enough to show the relationship between tree size and elevation, why they are presented in Fig.2? On the other hand: if this relationship was not aim of your study, why did you mention it for some species, and not for others?

L201-207: This entire paragraph is methods, and not results.

L281: You repeatedly mention that some of the results you describe and discuss here were not measured and/or aim of the study. I would avoid this because it makes the entire manuscript and findings a bit weak. If not aim of your study and/or not measured, simply excluding it from the text would be better.

Additional comments

In L79 you state that few studies exist in the Pacific NW. Which ones? Their main findings should be briefly described and adequately cited.

Again Fig.2: You state that tree dbh-elevation was not aim of your study. If so, you should delete Fig.2. If not, please justify the presentation of OHWM-dbh/basal area relationships through the entire text (starting in the last paragraphs of the Introduction).

All Figures should be self-explaining, without the need to read the text. Figures 2 and 3 use a lot of abbreviations for the single species which are not explained. Please explain all abbreviations in the legends.

Table 1: Avoid wording such as "generally perched" and "generally inundated". This makes the entire reasoning weak. It seems that youself are not convinced of the methodology and the delineation of the OHWM.

---

## Round 0.2 · Minor Revisions

Dear Nate Hough-Snee,

I have now received a review for your revised manuscript. I agree with the reviewer that the manuscript has been substantially improved. As you will see, these new comments are mostly about the form but I hope this additional review will help you in preparing a manuscript that will be easily read and understood. I'm looking forward reading your revised version.
Sincerely,

Yann Salmon



Minor comments:

In relation to the reviewer 2 original comment: “In the statistical analyses, I missed a species rarefaction and/or species-area curve to test for representativity of your vegetation/species samples. Given the sampled area, is your inventory representative for the different vegetation types in both the understory and overstory?
This was not done. The area was considered representative of the wetland.”
Please add this last point somewhere in the text it might be helpful for other readers.

Discussion: I agree with the reviewer that it might help the reader if you would more clearly state in the discussion where you are answering your two primary questions. The discussion represents about one third of the manuscript, which is quite substantial, and some readers might just be looking for the answer to one of those questions. If you do not want to follow the original reviewer suggestion to have sections, you could for example refer to question 1 and 2 in the text.

in Fig. 2.
The regression for C. ACMA, and particularly for E. COCO and H. HODI should probably not be presented (or at the very least, there should be a warning sign), because they are basically driven by one point only (at DBH 4 for the COCO and ACMA and 2.8 for HODI) and thus extremely sensitive to any error of that one single point. Of course a statistical software will still be able to make a regression, but it doesn’t mean it is valid. In those three cases, have you tried to look at the distribution of your data and at the residual of linear regression to see if the assumptions underlying the linear regression approach are met?
This uncertainty is kind of visible in the confidence interval (or whatever it is. The information about the shaded area is missing in the caption),

Fig. 3 please indicate in the caption what is shown by the boxplot (e.g., mean/median, what are upper and lower box limits, and whiskers)

Fig. 4 and 5, if you do not list the species code here, please refer to the part of the manuscript where it can be found (e.g. table 3)

Table 2 & 3. even if you don’t use all the wetland indicator terms in the manuscript, you need to explicit them in the caption since you present them in the table. Something short (e.g., what you did for FACW L322) is sufficient with the reference.

Appendix. These are actually supplemental information. Please update name according to PeerJ guideline:
“File naming format: 'Supplemental [Item] S[number]' e.g. 'Supplemental Data S1'.
Use the following style when citing Supplemental files in the text:
Fig. S1, Table S1, Data S1, Video S1, Article S1, Audio S1.”

·

Basic reporting

The author has improved the latest version of the paper. Well done! However the Discussion and Conclusion sections of the new version still lacks structure.
I had suggested before that the Discussion could be sub-divided to address each of the objectives. It will be helpful for the reader to understand the major findings of the paper if the author considered addressing his two objectives clearly. The author choose to not incorporate this in the revision by saying "This is a stylistic change, and while I appreciate the suggestion, the discussion is relatively short and transitions abound between sections."
I do not think that the Discussion is relatively short, there is still substantial information there but if the author wishes to follow his format then I'd suggest the following changes;
- re-write the first paragraph to clearly discuss that species assemblages represent elevation gradient across the wetland surface. You do not need to stress again what you did, “Here I quantified…” you have already covered this in your methods.
- Line 289 “high to low” what “within a wetland”?
- “This study” has been used repeatedly in the paragraph in line 358.
- Check sentence in lines 382-384. Define "designed hydrology" if you want to use the term here?

Experimental design

Methods described with sufficient detail

Validity of the findings

- Conclusion – sentence in lines 390-392 arbitrary and not necessary here. Expand the sentence in lines 392-394 to elaborate your findings which will address both your objectives

Additional comments

- Figure 1 –
(i). useful to include the names of some of the major streams within the catchment. Contour lines may also be useful to show elevation.
response from author - Within this figure there are no named streams
(ii). I still do not like how Figure 1 looks. Suggestion to make it a bit aesthetic – add in an insert map of the country showing Washington State then a box around Ash Wetland. Grids would be helpful too. That grey bit of the State in the legend does not look nice. Also you could make the bigger map satellite and insert map of the wetland topographic. Also Ash Wetland must have some kind of landmark – label that on the map.
- Figure 2 –
(i). graphs need to be made clearer. Could DistWater be changed to different symbols rather than a colour gradient? Increase font size of the axis labels
response from author - The colour gradient has been removed.
(ii). DistWater colour gradient still there. Increase the font size. Figure caption is self-explanatory now – well done
- Swap figures 2 and 3 as you mention figure 3 first in your results, might as well call it Figure 2 (line 213)
- Figure 3 box plot reads much better but I wonder if instead of having the species on the x-axis in alphabetical order they could be based on average elevation? so you can see a pattern from lowest to highest elevation.
- Figure 6 – plants species in photographs not labelled. Arrow on the side not useful
- Break the sentence in lines 113 – 115 in to two.
- Line 116 “Wetland has an average elevation of 281-meters” above what?
- Line 117/118 “The water table rises with fall and winter rain and falls throughout the growing season into late summer” – check this line. Also fall could just be called autumn like all the other seasons to avoid confusion with falls which I presume means ‘rainfall’
- Line 133/134 “These zones were based on within wetland elevation” – check sentence tense
- Careful with presentation of "P" – value, either have it italised throughout the document or not, e.g. 206, 229. Technically I think it should be lower case and not in italics (p = 0.0001)

---

## Round 0.3 · accepted · Accept

Just remember to add the units (m) in the abstract when you give the elevation (L37-38). Additionally, data is usually considered a plural in scientific writing (e.g. L47).